# Analysis of Retrospective Laboratory Data on the Burden of Bacterial Pathogens Isolated at the National Veterinary Research Institute Nigeria, 2018–2021

**DOI:** 10.3390/vetsci10080505

**Published:** 2023-08-05

**Authors:** Dennis Kabantiyok, Moses D. Gyang, Godwin O. Agada, Alice Ogundeji, Daniel Nyam, Uchechi G. Uhiara, Elmina Abiayi, Yakubu Dashe, Sati Ngulukun, Maryam Muhammad, Oyelola A. Adegboye, Theophilus I. Emeto

**Affiliations:** 1Diagnostic Laboratory Services Division, National Veterinary Research Institute NVRI, PMB 01, Vom 930010, Nigeria; 2Menzies School of Public Health, Charles Darwin University, Casuarina, NT 0811, Australia; 3Public Health & Tropical Medicine, College of Public Health, Medical and Veterinary Sciences, James Cook University, Townsville, QLD 4811, Australia; 4World Health Organization Collaborating Center for Vector-Borne and Neglected Tropical Diseases, James Cook University, Townsville, QLD 4811, Australia

**Keywords:** retrospective, antimicrobial resistance, farm animals, pathogens, Nigeria

## Abstract

**Simple Summary:**

Farm animals are important in food security and safety. The presence of disease-causing bacteria in farm animals is a key indicator of animal health and food safety. To understand the distribution and treatment outcome of these microorganisms in farm animals, data from the Bacteriology Laboratory of the National Veterinary Research Institute, Nigeria, from 2018 to 2021 were analysed. The influence of different types of antibiotic intervention from 2018 to 2021 on these bacteria was investigated. The findings showed that avian species were the predominant farm animals. The study revealed that the most isolated bacteria were *Escherichia coli*; *Salmonella* spp.; *Klebsiella* spp.; *Staphylococcus* spp.; *Proteus* spp.; and *Pseudomonas* spp. Additionally, it was noticed that the number of bacteria isolated increased steadily through the years, accompanied by the resistance to common antibiotics used during the period. Noteworthy is the resistance of *Klebsiella* spp. to cephalosporins—an important second-generation antibiotic. This is worrisome because the increasing resistance of animal bacteria portends more resistance to antimicrobial resistance in humans through several means. To address this growing concern prudent use of antimicrobials, effective surveillance, and stricter biosecurity measures (to minimise the need for antimicrobials) in farms must be observed to understand this problem better and minimise the spread of these infectious agents.

**Abstract:**

Farm animals harbour bacterial pathogens, which are often viewed as important indicators of animal health and determinants of food safety. To better understand the prevalence and inform treatment, we audited laboratory data at the Bacteriology Laboratory of the NVRI from 2018–2021. Antibiotics were classified into seven basic classes: quinolones, tetracyclines, beta-lactams, aminoglycosides, macrolides, nitrofuran, and cephalosporins. Trends were analysed using a generalised linear model with a log link function for the Poisson distribution, comparing proportions between years with an offset to account for the variability in the total number of organisms per year. Avian (73.18%) samples were higher than any other sample. The major isolates identified were *Escherichia. coli*, *Salmonella* spp., *Klebsiella* spp., *Staphylococcus* spp., *Proteus* spp., and *Pseudomonas* spp. We found that antimicrobial resistance to baseline antibiotics increased over the years. Of particular concern was the increasing resistance of *Klebsiella* spp. to cephalosporins, an important second-generation antibiotic. This finding underscores the importance of farm animals as reservoirs of pathogens harbouring antimicrobial resistance. Effective biosecurity, surveillance, and frugal use of antibiotics in farms are needed because the health of humans and animals is intricately connected.

## 1. Introduction

Microbe-host interactions can have either beneficial or pathogenic outcomes for both the microbe and the host. One of the major public health concerns related to these interactions is the presence of food-borne pathogens (viruses, bacteria, parasites, fungi) and toxins in farm animals and their products [1]. These pathogens and toxins not only affect the safety and quality of food but also pose a threat to human health through zoonotic transmission [2]. The ecology of microbes around farm animals and their products is influenced by several factors, such as biosecurity measures, microbial flora of livestock, hygiene practices, and environmental conditions [2]. 

Bacterial agents are predominant among the food-borne pathogens from animal sources; for example, *Salmonella* spp., *Staphylococcus aureus*, *Escherichia coli (E. coli)*, and *Listeria monocytogenes* [3,4]. Food animals serve as reservoirs of zoonotic pathogens, hence crucial in transmitting infection from food sources [1]. For example, *E. coli*, a commensal bacterium, does not normally cause disease in food animals. However, it has been documented as a promiscuous microbe that can acquire and disseminate plasmid-mediated antimicrobial resistance genes [5,6,7]. Similar phenomena have been observed in other microorganisms identified in food animals [8,9], such as *Brucella abortus*, *Clostridium difficile*, *Campylobacter jejuni*, *Enterococcus* spp., and *Aeromonas hydrophilia* [10,11]. 

In Nigeria, previous studies on animal products have isolated pathogenic microbes with zoonotic/public health potentials, such as methicillin-resistant *Staphylococcus aureus* (MRSA), *Campylobacter* spp., *Shigella* spp., and *Listeria monocytogenes* [11,12,13,14]. These pathogens can negatively impact farm animal health and production, resulting in economic losses. Extensive usage of antimicrobials in farm animals has compounded the deleterious effect of food-borne pathogens through the emergence of resistant microorganisms. A study by Van Boeckel et al. (2017) estimated that up to 73% of antimicrobials manufactured globally are used in farm animals [12], leading to environmental contamination by residual antimicrobials and the eventual buildup of more resistance [13]. The seriousness of antimicrobial resistance in veterinary practice has brought about efforts by regional groups such as the European Medicines Agency’s European Surveillance of Veterinary Consumption (ESVAC), the Canadian Integrated Program for Antimicrobial Resistance Surveillance (CIPARS), and the Japanese Veterinary Antimicrobial Monitoring System (JVARM) [14]. The situation of antimicrobial resistance (AMR) in Africa is complicated by several factors, including lack of access to appropriate antimicrobials, poor legislation on antimicrobial use, weak surveillance systems, and insufficient knowledge of treatment guidelines [15,16].

Food-borne pathogens are a major public health concern worldwide, causing up to 3 million deaths annually [17]. Global food-borne infections of bacterial origin in 2015 were estimated at 226,526,634 [18], with food sourced from animals as a major contributor to these infections [19]. This data is expected to rise with the increasing human population. However, this estimate may not reflect the true burden of the problem, as many cases are underreported in developing countries, including Nigeria [20,21]. As the global population and the demand for animal protein increases, so does the need for food safety and security. The livestock industry in Nigeria contributes between 6 and 8% of the country’s GDP and employs many smallholder farmers and nomads [22]. The poultry sector alone has an estimated number of 180 million birds, ranking second in Africa after South Africa [23]. 

It is projected that the poultry population in Nigeria will grow to 1.3 billion heads by 2050 [24]. Therefore, health authorities must implement measures that ensure safer food production and consumption for the expanding population and create wealth through safe food. This data audit covers a period of four years and aims to provide a snapshot of the microorganisms isolated at the bacteriology laboratory of the National Veterinary Research Institute (NVRI), Vom, Nigeria. It also seeks to understand these pathogens’ distribution and their antimicrobial resistance.

## 2. Materials and Methods

### 2.1. Study Area and Data Collection

The National Veterinary Research Institute (NVRI) is a leading research institute in Nigeria that researches animal diseases, diagnosis, training, and production of biologicals for the livestock industry. The NVRI operates through six zonal stations and 17 laboratories across Nigeria [25]. One of these laboratories is the bacteriology laboratory at the Central Diagnostic Laboratory, located at the headquarters of the NVRI (Figure 1). This laboratory performs bacteriological investigations of specimens received from various sources, mainly from the Plateau State and its neighbouring regions. A large proportion of the specimens are submitted for postmortem examination to determine the probable cause of death in livestock. Each specimen is assigned a unique identification identifier that is linked to additional information such as farm location, flock size, and number of deaths. A flow chart of the data generated for this study is shown in Figure 2.

### 2.2. Key Pathogens

#### 2.2.1. *Escherichia coli*

*Escherichia coli* is a bacterium that colonises the gut of all vertebrate animals as either a commensal or an opportunistic microbe. It provides digestive support by synthesising certain vitamins and essential biomolecules [26,27]. However, it can also cause various infections in both human and animal hosts, such as colibacillosis, hospital-acquired pneumonia, haemolytic uremic syndrome (HUS), meningitis, and urinary tract infection (UTI) [26,27]. Researchers often classify pathogenic strains of *E. coli* into pathotypes using acronyms that define the target organ, a virulence gene, or a specific gene such as extra-intestinal pathogenic *E. coli* (ExPEC), avian pathogenic *E. coli* (APEC), and Shiga toxin-producing *E. coli* (STEC) [28]. Some *E. coli* strains may carry plasmids that facilitate their ability to acquire and distribute antimicrobial resistance horizontally to other microbes or the environment. Previous studies have shown that APEC strains can serve as reservoirs of virulence genes for ExPEC [29], especially in poultry [30]. Because of their ubiquitous nature in mammals, *E. coli* is a useful indicator for monitoring antimicrobial resistance, as the frequency of antimicrobial resistance by *E. coli* can provide useful insight into selective pressure induced by drugs [31,32].

#### 2.2.2. *Staphylococcus* spp.

*Staphylococcus* spp. is a group of bacteria that cause various diseases in humans and animals. These bacteria can infect the skin and mucous membranes, causing conditions such as otitis, pyoderma, and surgical wound infections [32]. Among the challenges posed by *Staphylococci* spp. is their ability to develop multidrug resistance, especially in *Staphylococcus aureus*, which is a common cause of hospital- and community-acquired infections and food poisoning [33,34]. *Staphylococcus aureus* belongs to the ESKAPE (*Enterococcus faecium*, *Staphylococcus aureus*, *Klebsiella pneumoniae*, *Pseudomonas aeruginosa,* and *Enterobacter* spp.) group of the pathogen which have been given a ‘priority status’ [35] in the fight against antimicrobial resistance (AMR) [36]. Moreover, *Staphylococcus aureus* can colonise or infect healthy, warm-blooded animals, increasing the risk of transmission to humans [37]. Knowledge of diseases caused by *Staphylococcus aureus* in animals may vary depending on the economic impact of the disease and the life expectancy of the animal. For instance, mastitis, which affects milk production and calf nutrition-with economic consequences on agribusinesses, gets more attention than osteomyelitis or endocarditis, which are often not noticeable because their progression outlives most feed animals [37].

Although the coagulase-negative group of *Staphylococcus* (CoNS) was previously considered less virulent than *Staphylococcus aureus* [38], recent studies have revealed that they possess a range of virulence factors, such as exfoliative toxins, hemolysins, enterotoxins, and toxic shock syndrome toxin [38,39,40,41]. CoNS have been associated with mastitis in animals and biofilm formation on biotic and abiotic surfaces, which facilitates colonisation and infection by other microbes [41].

#### 2.2.3. *Salmonella* spp.

Salmonella is a genus of gram-negative, facultatively anaerobic, flagellated bacilli belonging to the family of Enterobacteriaceae [42]. The surface antigen of the cell’s lipopolysaccharide, flagella, and capsular polysaccharide is expressed as the O, H, and V antigens and is used to classify its serotypes [43]. Salmonella bacteria are ubiquitous and resilient in nature and can survive for several weeks in dry environments and several months in water [43]. The genus is currently divided into *Salmonella bongori* and *Salmonella enterica*, with a third species pending approval (*Salmonella subterranean*). *Salmonella enterica* has more than 2610 serovars that cause diseases in humans and animals [44]. This species is further divided into six subspecies: *Salmonella enterica enterica*, *Salmonella enterica salamae*, *Salmonella enterica arizonae*, *Salmonella enterica diarizonae*, *Salmonella enterica houtenae,* and *Salmonella enterica indica*. Two serovars of *Salmonella enterica*, *Salmonella gallinarum* and *Salmonella pollurum*, are the most commonly associated with clinical disease in poultry [44].

*Salmonella* is a bacteria pathogen that causes infectious diseases in humans and animals, with chickens being one of its main hosts [45]. It can cause chronic and acute diseases in poultry [46] and swine [47,48], resulting in serious economic losses for farmers and requiring large investments of private and public resources for testing and control [49]. In addition, salmonellosis can contaminate human food, posing a significant threat to public health [50,51]. 

#### 2.2.4. *Klebsiella* spp.

*Klebsiella* spp. are a group of gram-negative, non-motile, lactose fermenting, facultative anaerobic negative rods belonging to the Enterobacteriaceae group [52]. The *Klebsiella* genus consists of various species, such as the *Klebsiella pneumoniae* species complex (KpSC) and several other genetically distinct species [53], which encompasses a broad range of species, including *K. indica*, *K. terrigena*, *K. spallanzanii*, *K. huaxiensis*, *K. oxytoca*, *K. grimontii*, *K. pasteurii*, and *K. michiganensis* [54]. These species share an average nucleotide identity of only 90% with KpSC. They are ubiquitous in the environment and are found in soil, water, manure, and plants; as such, they have been isolated from humans and animals alike [52,55]. 

### 2.3. Laboratory Investigation 

The study employed standard laboratory procedures to isolate bacterial pathogens from the samples. These procedures involved basic cultural techniques, biochemical fermentation tests, and morphological observation of the isolates [56]. The samples were simultaneously inoculated on MacConkey and blood agar, except for those that required a pre-enrichment medium for specific investigations. Antimicrobial susceptibility testing (AST) was conducted using the Kirby-Bauer disc diffusion method [57] for all the identified bacterial isolates using validated in-house discs [58] impregnated with various classes of antibiotics. To facilitate data analysis, we classified the antibiotics into seven groups based on their primary classes, namely, quinolones, tetracyclines, beta-lactams, aminoglycosides, macrolides, nitrofuran, and cephalosporins. This classification was necessary because of the differences in the trademark names used over time. 

### 2.4. Data Analysis

All data analysis and visualisation were conducted in R, version 4.1.1 (R Core Team, 2021). Trends were analysed using a generalised linear model with a log link function for the Poisson distribution, comparing proportions between years with an offset to account for the variability in the total number of organisms per year. Percentage increases and 95% CIs were estimated to determine significant changes in the relative proportion of bacterial pathogens.

## 3. Results

A total of 2362 tests were conducted between 2018 and 2021, with avian samples accounting for the majority (73.18%), followed by bovine and canine samples (5.67% and 5.27%, respectively), while Feline, Equine, and Pisces samples were the least. There was a yearly increase in samples except for 2020 (Table 1).

Poisson regression analysis showed that *Klebsiella* spp. and *Proteus* spp. were the only isolates that demonstrated a significant increase in their number of isolates between 2019 and 2020. All other isolates demonstrated a decrease in their number, with *E. coli* showing a similar number of isolates compared to the previous year (Table 2). In contrast, all the isolates increased by at least two-fold between 2020 and 2021. The major pathogens isolated are *E. coli*, *Staphylococcus* spp., *Proteus* spp., *Salmonella* spp., *Pseudomonas* spp., and *Klebsiella* spp. (Figure 3), which are the leading pathogens. Regardless of the dip in 2020, there was a general increase in isolates from 2018–2021.

The antimicrobial resistance of the selected isolates to selected antibiotics showed that *Escherichia coli* and *Salmonella* spp. exhibited higher resistance to tetracyclines and macrolides across all samples (Figure 4). Whereas *Klebsiella* spp. alone showed susceptibility to nitrofurans and aminoglycosides, cephalosporins and aminoglycosides were relatively effective against most pathogens.

A majority of the samples originated from Plateau State, Nigeria, as depicted in Figure 5. The number of samples increased alongside their distribution; Bauchi and Kogi were the leading states after Plateau in 2021. In 2020, there was an appreciable reduction in the number of cases and isolates compared to previous years. While in 2021, there was a significant rise in both cases and isolates, with a more extensive geographical reach covering multiple states.

## 4. Discussion

Our study highlights the consistent increase of antimicrobial-resistant pathogens in food animals. Furthermore, the most common pathogens identified (*Escherichia coli*, *Staphylococcus aureus*, *Salmonella* spp., *Pseudomonas* spp., and *Klebsiella* spp.) showed high levels of resistance to common classes of antibiotics except for aminoglycosides and cephalosporins, which is alarming. Resistance to these antibiotics is alarming due to their crucial importance as first-line therapeutics [59] and the role of animal reservoirs of antimicrobial resistance in the spread and emergence of resistant species in the environment [60,61]. While resistance to quinolones, macrolides, tetracyclines, and beta-lactams was common among all isolates, *Klebsiella* spp. was an exception, as it was only susceptible to nitrofuran and aminoglycoside. The data from this study showed that *Klebsiella* spp. had developed significant resistance to this drug—an important second-generation antimicrobial often used for treating veterinary infections. This is corroborated by a study by De Oliveira et al. (2020), who reported that *Klebsiella* spp. resistance to cephalosporins is a global phenomenon [36]. 

The analysis of the samples revealed the dominant pathogens and their AMR profiles, which have important implications for food safety, public health, and business. Additionally, most of the samples (62%) were from poultry, which reflects the growing popularity and importance of poultry farming in Nigeria [22]. However, this also means that the data were skewed towards avian species, which might limit the generalisability and accuracy of the findings for other animal pathogens. Therefore, future studies in settings with other species are needed to obtain a more comprehensive picture of the AMR situation in farm animals.

### 4.1. AMR Spilling from Animal to Humans

Antimicrobial resistance has been described as a slow-pandemic, with estimated deaths from antimicrobial-related complications projected to reach up to 10 million by the year 2050 [31]. The contribution of livestock to global antimicrobial resistance is significantly documented [62], and it has been estimated that about 73% of global antimicrobial use is in livestock [63]. The US alone consumes an estimated 10.9 million kg of antibiotics in domestic livestock production systems [64]. Thus, livestock play an important role in the dissemination of antimicrobial resistance, and they do this through a variety of means: food value chain [65], direct contact with farm owners and workers, and byproducts such as manure [66,67]. This means it is imperative to keep up with data on antimicrobial resistance in animals. Unfortunately, data on antimicrobial use in low- and middle-income countries(LMIC) is inadequate because only 6% of countries have a system that monitors antibiotic resistance in animals [68]. And then there are other factors that complicate AMR in LMIC, such as poor regulation/control, irrational use to make up for poor animal husbandry, and biosecurity [69]. The presence of resistant bacteria in animal reservoirs, particularly those carrying resistance to antimicrobial agents that are critically important for human therapy, such as aminoglycosides, fluoroquinolones, and third- and fourth-generation cephalosporins, poses a serious threat to public health [70]. There are several complex scenarios where AMR from livestock is capable of getting to humans. In a study by Nikola et al. (2020) [71], the R plasmids typical to *E. coli* from birds fed with tetracycline-supplemented feed were isolated from the human faeces of individuals who have worked on the same farm. A variety of health conditions in humans has been connected to the effect of antimicrobial usage, including bone marrow toxicity, hepatotoxicity, carcinogenicity, and immunological and reproductive disorders [72]. Hence, it is relatable to say that man is already living with the consequences of antimicrobial resistance. This view is supported by the fact that the rising antimicrobial resistance in animals is often a sign of excessive use or abuse, which can have indirect negative consequences on human health. Although it is difficult to definitively link these negative consequences to the excessive use of antimicrobials, the deteriorating health condition in developing countries may be compounded by this practice. Therefore, it can be argued that the burden of antimicrobial resistance in humans and animals is intricately connected. 

### 4.2. The Influence of the COVID-19 Pandemic on the Livestock Industry

The majority of the samples in this study were from Plateau State, Nigeria, which may be attributable to the proximity of the laboratory to immediate communities. There was a steady increase in both isolates and samples over the years, except for a noticeable drop in 2020. This decline might be attributed to the COVID-19 pandemic and lockdown, which affected the agribusiness and livelihood of farmers. This finding is consistent with previous studies that reported the negative impacts of the pandemic on the agricultural sector [73,74]. However, there was a significant rise in both cases and isolates, with a more extensive geographical reach covering multiple states in 2021. This was possibly due to the return to normalcy following the easing of restrictions during COVID.

The most common pathogens isolated from the samples were *Escherichia coli*, *Staphylococcus aureus*, *Salmonella* spp., *Pseudomonas* spp., and *Klebsiella* spp. The diversity and frequency of these bacteria increased steadily from 2018 to 2021, and except for a slight decline in 2020, the percentage yearly increase of isolates also followed a similar trend. Notably, all the members of the ESKAPE group (*Enterococcus*, *Staphylococcus*, *Klebsiella*, *Pseudomonas*, and *Enterobacter*) were represented in the isolates, with *Staphylococcus* being the most prevalent, followed by *Klebsiella* and *Pseudomonas*. These bacteria are known to contribute to the emergence and spread of AMR through their acquisition and distribution of resistance genes through genetic mutation and mobile genetic elements [35,75]. Moreover, the presence of *Salmonella* and *Staphylococcus* spp. portends a significant risk to food safety, public health, and agribusinesses as potential carriers of food-borne infections and intoxications [76,77,78].

The spatial distribution of samples shows a strong clustering around Plateau State (likely because the NVRI is domiciled in Plateau State), with few cases reported from the neighbouring states/regions. Nonetheless, a greater diversity of cases was observed in 2021 compared to previous years. It should be noted that the data presented in this study are limited to the cases handled by the bacteriology laboratory in Plateau State and do not represent an exhaustive account of all the cases treated by state and zonal branches.

### 4.3. Veterinary and Food Safety Implications of Bacterial Pathogens on Public Health

*Klebsiella* infections in humans are often described as a hospital-acquired infection prevalent in immunocompromised patients. Infections outside the clinical settings involve diverse routes, such as ingesting contaminated food and water and direct contact with companion animals [79]. Common transmission routes in livestock and companion animals include consuming contaminated food and water and through the genitourinary and intramammary pathways [80]. Although most infections with *Klebsiella* go unnoticed, pathogenic *Klebsiella* infections in humans produce harmful effects that are sometimes fatal, such as blood infection, respiratory diseases, abscesses, and encephalitis [81].

*Klebsiella* spp. acquisition of genetic elements and its ability to initiate mutations that provide resistance to antimicrobials is a formidable trait associated with its virulence. This can lead to the emergence of convergent clones known as multidrug-resistant and hypervirulent (MDR-hv) *Klebsiella* spp. [82] These hypervirulent strains continue to proliferate due to limited treatment options, thereby setting the stage for the emergence of multidrug resistance (MDR) superbugs with dire public health consequences [83]. This is not surprising, seeing the level of antimicrobial use in feed and companion animals.

*E. coli* is a bacterium that causes various diseases in farm and companion animals, posing significant challenges to veterinary practice and food safety. For example, avian pathogenic APEC causes colibacillosis in poultry, leading to serious respiratory distress [84]. Enterotoxigenic *Escherichia coli* (ETEC) produces enterotoxins that result in diarrhea in pigs and calves, often with fatal outcomes [32]. Moreover, *E. coli* is associated with over 80% of mastitis cases in cattle, affecting the quality of milk and the nutrition of young calves. *E. coli* often co-infects with other pathogens such as *Staphylococcus aureus*, *Streptococcus agalactiae*, and *Streptococcus uberis* [32]. The emergence of extended-spectrum beta-lactamase (ESBL) *Escherichia coli* is of great concern to veterinarians, as most of the antibiotics approved for veterinary use belong to the class of beta-lactams and cephalosporins. The high rate of antimicrobial resistance acquisition and dissemination by *Escherichia coli* reduces the effectiveness of treatment options [31].

*Staphylococcus* spp. has been isolated from various animal species, but they cause the most severe diseases in ruminants. Cattle are considered a major reservoir of *Staphylococcus aureus*, which is the most prevalent species among different climates and animals. However, swine have also emerged as an important reservoir of methicillin-resistant *Staphylococcus aureus* (MRSA), a strain that seriously threatens human health [85]. A specific MRSA strain, ST398, has been found in swine, humans, and birds [86]. The contact between humans and animals, such as farm owners and veterinarians, facilitates the transmission and spread of MRSA in the community. The extent of diseases caused by *Staphylococcus* spp. in animals may depend on the economic impact of the disease and the life expectancy of the animal. For example, mastitis, which affects milk production and calf nutrition, receives more attention than osteomyelitis or endocarditis, which may not be noticeable in feed animals [37].

Salmonellosis is a major food-borne disease affecting millions worldwide [87]. It can be classified into typhoidal and non-typhoidal forms, depending on the serotype of *Salmonella* bacteria involved. Both forms are endemic in many developing countries and are mainly transmitted through food and water contaminated with fecal matter. According to global estimates, non-typhoidal salmonellosis causes about 93.8 million cases of gastroenteritis [87] and 155,000 deaths annually [88], especially among vulnerable groups such as infants and elderly persons. The most common serotypes associated with human salmonellosis are *Salmonella enteritidis* and *Salmonella typhimurium* [89]. *Salmonella* bacteria can infect a wide range of hosts, including humans, reptiles, rodents, birds, and amphibians. Therefore, human salmonellosis can result from direct or indirect contact with infected animals or their products [90]. Poultry products, particularly undercooked meat and raw eggs, are the main sources of human salmonellosis [91]. However, the risk of *Salmonella* infection also depends on other factors, such as lifestyle, eating habits, and environmental conditions. People living in low-income settings with poor hygiene and limited access to safe water and food are more susceptible to *Salmonella* infection [75].

## 5. Conclusions

Food animals can harbour diverse bacterial pathogens that are resistant to antimicrobial agents critically important in human therapy. Through constant surveillance, monitoring for changes in their antimicrobial resistance pattern may intimate appropriate action. A combination of approaches, such as improved biosecurity at farms, the frugal use of antimicrobial agents, and a surveillance system for monitoring and reporting antimicrobial resistance in farms, hold the promise of mitigating antimicrobial resistance. As such, there is a need to heighten AMR surveillance in animal pathogens with the intention of developing baseline AMR data in animals due to their potential impact on public and human health.

## Figures and Tables

**Figure 1 vetsci-10-00505-f001:**
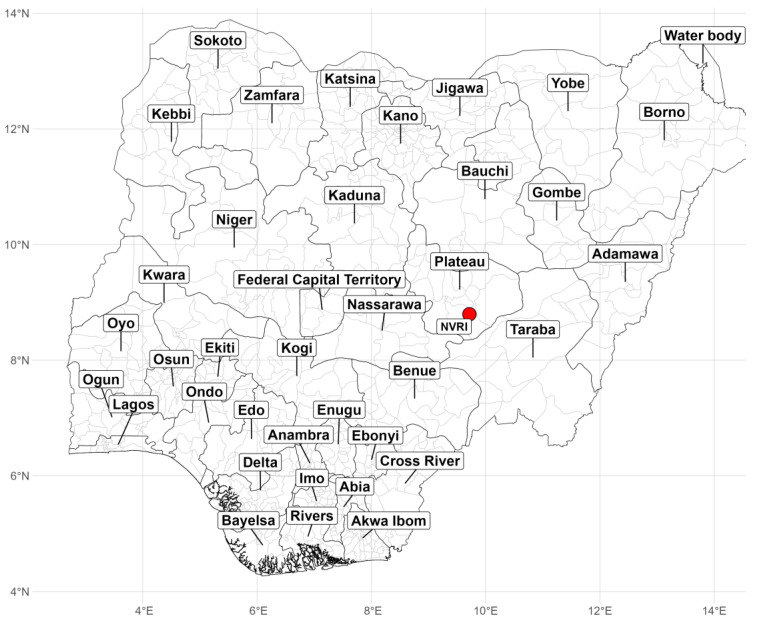
The map of Nigeria showing the location of the NVRI, Vom, Plateau State. The red dot represents the location of NVRI.

**Figure 2 vetsci-10-00505-f002:**
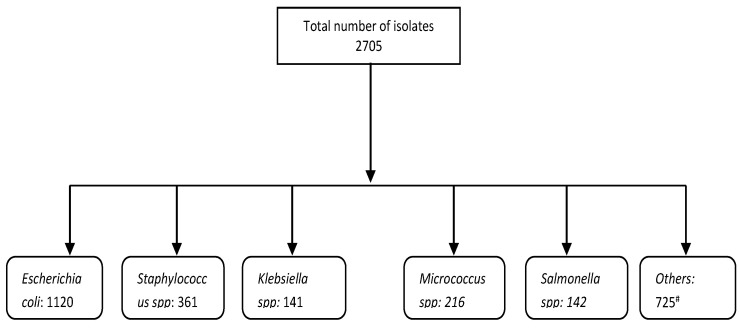
Flowchart of data. Others#: *Listeria* spp., *Citrobacter* spp., *Bacillus* spp., *Aeromonas* spp., *Pseudomonas* spp., *Enterobacter* spp., *Shigella* spp., *Streptococcus* spp., *Yersinia* spp., *Edwardsiella* spp., *Leptospira* spp., Bacteroides.

**Figure 3 vetsci-10-00505-f003:**
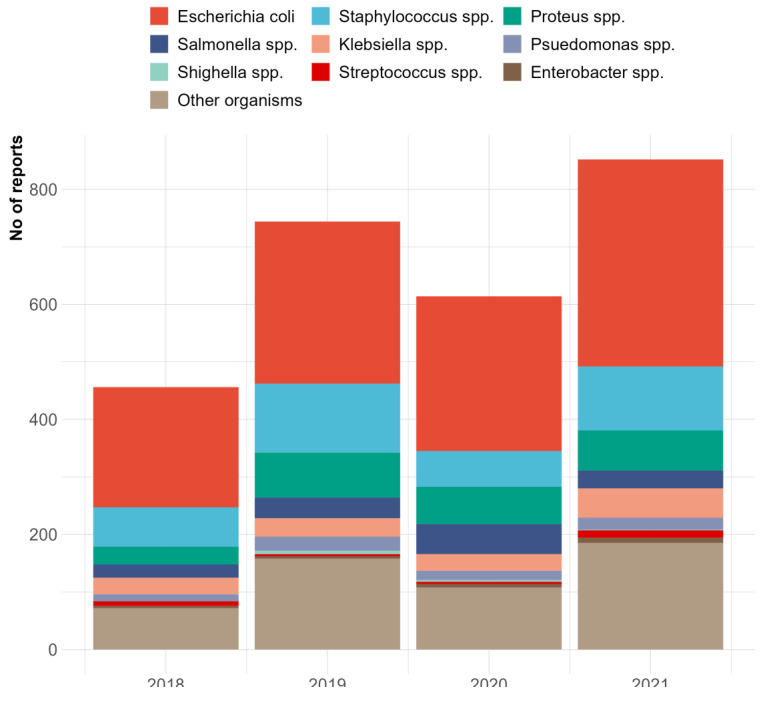
Cases of bacterial isolates identified at the NVRI bacteriology laboratory between 2018 and 2021.

**Figure 4 vetsci-10-00505-f004:**
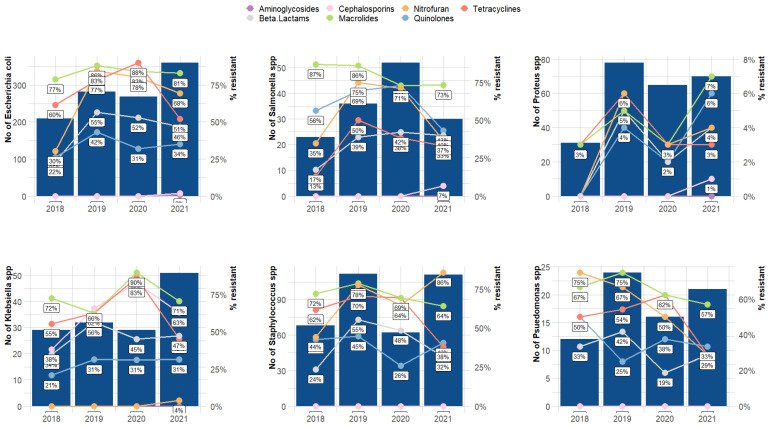
Antibiotic resistant trends of the antibiotics. The bars represent the number of selected isolates, while the lines show the percentage of antimicrobial resistance to specific antibiotics.

**Figure 5 vetsci-10-00505-f005:**
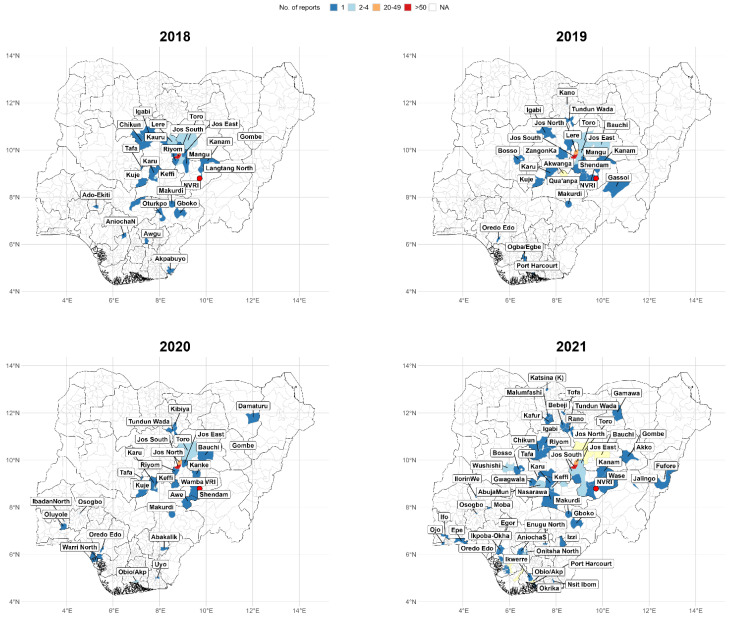
Geographical distribution of samples reported to NVRI from 2018–2021.

**Table 1 vetsci-10-00505-t001:** Descriptive summary of the sources of the sample.

Sample	2018	2019	2020	2021
Avian	310	383	323	468
Bovine	29	33	24	29
Canine	19	24	27	27
Caprine	8	11	11	15
Equine	1	1	0	7
Feed	1	4	1	0
Feline	1	0	0	3
Laprine	22	23	21	27
Ovine	10	10	13	16
Pisces	2	0	2	4
Porcine	6	9	32	58
Grand Total	409	498	464	654

**Table 2 vetsci-10-00505-t002:** Percentage yearly increases of isolates between the years 2018–2021.

Microorganisms	% Bacterial Isolates for Each Animal										Number of Reports 2018–2021			% Annual Increase (95% CI)
	Av	Bov	Can	Cap	Equ	Fel	Lap	Ov	Pcs	Por	Total	Mean	SD	
*Escherichia coli*	54	31	30	36	33	0	40	49	25	41	1120	280	62.09	15.83 (0.54, 21.13)
*Salmonella* spp.	9.5	0.9	0	4.4	22	0	0	0	25	1	142	35.5	12.23	11.31 (−3.44, 26.19)
*Klebsiella* spp.	4.5	8.7	5	2.2	0	25	8	14	0	8	141	35.3	10.59	18.04 (3.16, 33.14)
*Proteus* spp.	10	10	16	2.2	0	0	11	12	0	17	244	61	20.7	17.19 (5.88, 28.63)
*Staphylococcus* spp.	13	18	24	22	22	2	15	12	25	22	361	90.3	29.49	7.88 (−1.36, 17.16)
*Streptococcus* spp.	0.6	5.2	1	2.2	0	0	0	0	0	1	29	7.25	4.27	20.95 (−11.83, 55.01)
*Psuedomonas* spp.	2.4	5.2	3	4.4	0	0	3.2	4.1	0	4	73	18.3	5.32	10.44 (−10.11, 31.24)
*Shighella* spp.	0.5	0	1	0	0	0	0	0	0	2	10	2.5	2.65	0.00 (−56.64, 56.64)
*Enterobacter* spp.	0.9	0.9	0	2.2	0	0	1.1	2	0	0	23	5.75	2.36	30.33 (−6.84, 69.89)
Other organisms											562	141	54.07	30.33 (−6.84, 69.89)

Av.; Avian, Bov.; Bovine, Can.; Canine, Cap.; Caprine, Fel.; Feline, Lap.; Laprine, Ov.; Ovine, Pisces.; Pisces, Por.; Porcine.

## Data Availability

Data used for this study are unavailable to the public due ethical and privacy policy of the source.

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
