# Peer review of "Analysis of Retrospective Laboratory Data on the Burden of Bacterial Pathogens Isolated at the National Veterinary Research Institute Nigeria, 2018–2021"

_vetsci, 2023, doi:10.3390/vetsci10080505_

Round 1

Reviewer 1 Report

The authors in this manuscript have scientific merit that warrants publication in Veterinary Science (Analysis of Retrospective Laboratory Data on the Burden of Bacterial Pathogens Isolated at National Veterinary Research Institute Nigeria, 2018-2021). The authors have pointed out firstly the Retrospective Laboratory Data on the Burden of Bacterial Pathogens Isolated at Veterinary Research from different animals’ meat and in addition the alarming emergence of multi-drug resistance in different animal meat in Nigeria. Overall, the study undertaken by the authors is relevant and significant to the importance of the role of the veterinarian in the prevention and control of these food-borne pathogens in these variant reproductive animal farms in various cities in Nigeria. These will help the veterinarian in optimizing guidelines for using antibiotics. However, the manuscript needs to be rewritten for almost a section of material and methods, and the scientific names of bacterial genera and species should be printed in italics as well as authors have to present data in better ways in the table and Figure. In addition, it would be better to have a native English speaker review the English language of this manuscript.

Major and Minor comments:

Simple Summary and Abstract

·         Lines 19, 20, 21; Try to avoid we and our, and rewrite the sentence in a formal, academic style using the passive voice.  

1. Introduction:

·         Note: In all MS; you have to Italicize all bacterial names (genus and species)

·         Line 50: add “are” after These pathogens and toxins are not instead of These pathogens and toxins not

·         Lines 52, 54, 59, 58 and …….: space before [2] ; transmission [2] instead of transmission[2]

·         Lines 58,65: In all MS; you have to Italicize all bacterial names (genus and species)

·         Line 58.  E.Coli; it is preferable to rephrase sentences to avoid starting with abbreviations.

·         Line 61: give some examples in other microorganisms such

·         Line 68-70: you mentioned studies have shown but you only cited one reference. Change the sentence or add another reference,

·         Lines 75: No Abbreviations in the first time and write the complete name “The situation of AMR”.

·         Line 81: write estimation instead of number “226,526,634”.

·          

2. Materials and Methods

2.1. Study area and data collection

·         Line 111-112: Figure 2. Flowchart of data. This figure is similar to the result should move to the result section

·         Line 113-115; you have to Italicize all bacterial names (genus and species)

Others#: Listeria spp, Citrobacter spp, Bacillus spp, Aeromonas spp, Pseudomonas spp, Enterobacter spp, Shigella spp, Streptococcus spp, Yersinia spp, Edwardsiella spp, Leptospira spp, Bacteroides

2.2. Key Pathogens

 This section should shorten and move the introduction and the sentences related to the discussion part to the discussion section.

Escherichia coli

·         Lines 136; remove the reference and No Abbreviations in the first time and write the complete name ETEC, it preferable to rephrase sentences

Staphylococcus spp

Salmonella spp

Klebsiella spp

2.3 Laboratory investigation

·         Lines 236-247; it would better to cited both the protocol of bacterial identifications and the Antimicrobial susceptibility testing (AST) method

 2. Results

·         Lines 256-259 and 268: It would be better if this paragraph with “Table 1. Descriptive summary of the sources of the sample” move to 2.1. Study area and data collection

·         Line 270: Revise the title of Table 2. Percentage yearly increases of isolates between years from 2018-2021. 270, in addition, the number and % of bacteria for each animal should be present in this table.

·         Lines 273; in Figure; would be better if they represent data in a different way such as bacteria on the left side not on top in addition maybe with a bacterial number, and % of each bacteria in each year represent as well.

·         Lines 275-282: Revise the paragraph and avoid repetition from the material methods; remove   s(E. coli, Salmonella spp., Klebsiella 2spp., Staphylococcus spp., Proteus spp., and Pseudomonas spp.) to seven classes of antibiotics (tetracyclines, aminoglycosides, macrolides, beta-lactams, nitrofuran, quinolones, and cephalosporins)

·         Line 284 in Figure 4 removes the following (Tetracyclines, Aminoglycosides, Macrolides, Beta-Lactams, Nitrofuran, Quinolones, and Cephalosporins) to Escherichia coli, Salmonella spp, Proteus spp, Klebsiella spp, Staphylococci spp, and Pseudomonas spp.

·         Lines 287-293: Revise the paragraph and move into the discussion section “The influence of the COVID-19 pandemic on the livestock industry”, line 318

·         Line 295” in Figure would be better if add the cities name on Figures

4. Discussion

·         Lines 297-301: Revise the paragraph and avoid repetition from the sentences from the Result section

·         Lines 304-308: Revise the sentence and avoid repetition from the sentences from the Result section

·         Line 336; you have to Italicize all bacterial names (genus and species)

Good Luck

Dear Authors

MS is written in a simple and easy-to-understand manner. However, 

moderate editing of the English language is required. In addition, it would be better to have a native English speaker review the English language of this manuscript.

Best Wishes

Good Luck

Author Response

The authors thank the reviewer for their insightful comments. We have incorporated changes to address the concerns raised and hope all the issues have been addressed. Here is a point-by-point response to the reviewers’ comments below.

Reviewer 1:

Comments and Suggestions for Authors

The authors in this manuscript have scientific merit that warrants publication in Veterinary Science (Analysis of Retrospective Laboratory Data on the Burden of Bacterial Pathogens Isolated at National Veterinary Research Institute Nigeria, 2018-2021). The authors have pointed out firstly the Retrospective Laboratory Data on the Burden of Bacterial Pathogens Isolated at Veterinary Research from different animals’ meat and in addition the alarming emergence of multi-drug resistance in different animal meat in Nigeria. Overall, the study undertaken by the authors is relevant and significant to the importance of the role of the veterinarian in the prevention and control of these food-borne pathogens in these variant reproductive animal farms in various cities in Nigeria. These will help the veterinarian in optimizing guidelines for using antibiotics. However, the manuscript needs to be rewritten for almost a section of material and methods, and the scientific names of bacterial genera and species should be printed in italics as well as authors have to present data in better ways in the table and Figure. In addition, it would be better to have a native English speaker review the English language of this manuscript.

Response: Thank you for your detailed review and for pointing out key areas that must be reworked to improve the manuscript.

Comment1: Lines 19, 20, 21; Try to avoid we and our, and rewrite the sentence in a formal, academic style using the passive voice. 

Response: Thank you for your observation, however, veterinary sciences (as well as many academic journals) does not have a strict formatting style, hence the use of first person point of view. We have revamped the entire summary considering this suggestion.

Comment2: Note: In all MS; you have to Italicize all bacterial names (genus and species)

Response: We have taken note of this error. All microorganisms have been spelt correctly and italicized.

Comment3: Line 50: add “are” after These pathogens and toxins are not instead of These pathogens and toxins not

Response: Thank you for your suggestion, however, the sentence “These pathogens and toxins not only affect the safety and quality of food but also pose a threat to human health through zoonotic transmission.” is grammatically correct.

Comment4: Lines 52, 54, 59, 58 and …….: space before [2] ; transmission [2] instead of transmission[2]

Response: Thank you for your observation. We have corrected it and ensured all such mistakes in the manuscript are fixed.

Comment5: Lines 58,65: In all MS; you have to Italicize all bacterial names (genus and species)

Response: These have been corrected. We believe the response to comment #2 has also addressed all similar concerns.

Comment6: Line 58. E.Coli; it is preferable to rephrase sentences to avoid starting with abbreviations.

Response: We have made corrections to improve the sentence. The rephrased sentence now reads:

[For example, E. coli, a commensal bacterium does not normally cause disease in food animals. However, it has been…]

Comment7: Line 61: give some examples in other microorganisms such

Response: Agreed, we have included some additional microorganisms as advised.

Comment8: Line 68-70: you mentioned studies have shown but you only cited one reference. Change the sentence or add another reference,

Response: We have reworded the sentence accordingly.

Comment9:  Lines 75: No Abbreviations in the first time and write the complete name “The situation of AMR”.

Response: We have corrected this error.

Comment10: Line 81: write estimation instead of number “226,526,634”.

Response: Thank you for that observation.

Comment11: Line 111-112: Figure 2. Flowchart of data. This figure is similar to the result should move to the result section

Response: The flowchart represents the data extracted, we will like to keep it under section 2.1

Comment12: Line 113-115; you have to Italicize all bacterial names (genus and species)

Response: We have italicized the names (genus and species).

Comment13: This section should shorten and move the introduction and the sentences related to the discussion part to the discussion section.

Response: We have shortened the section and moved relevant materials to the discussion.

Comment14:  Lines 136; remove the reference and No Abbreviations in the first time and write the complete name ETEC, it preferable to rephrase sentences

Response: We agree and have revised it accordingly.

Comment15: Lines 236-247; it would better to cited both the protocol of bacterial identifications and the Antimicrobial susceptibility testing (AST) method

Response: We agree and have revised it accordingly.

Comment16:  Lines 256-259 and 268: It would be better if this paragraph with “Table 1. Descriptive summary of the sources of the sample” move to 2.1. Study area and data collection

Response: Thank you for that suggestion. However, we would like to retain the section and Table 1 as it is for coherence.

Comment17: Line 270: Revise the title of Table 2. Percentage yearly increases of isolates between years from 2018-2021. 270, in addition, the number and % of bacteria for each animal should be present in this table.

Response: We have reviewed the table and included the bacterial count for each animal.

Comment18: Lines 273; in Figure; would be better if they represent data in a different way such as bacteria on the left side not on top in addition maybe with a bacterial number, and % of each bacteria in each year represent as well.

Response: Thank you for your suggestion. We have adjusted the graph. It shows the number of selected isolates (bars) and the percentage of antimicrobial resistance to specific antibiotics (as lines). Adding the percentage of each bacteria will have too much information, which is distortion.

Comment19: Lines 275-282: Revise the paragraph and avoid repetition from the material methods; remove   s(E. coli, Salmonella spp., Klebsiella 2spp., Staphylococcus spp., Proteus spp., and Pseudomonas spp.) to seven classes of antibiotics (tetracyclines, aminoglycosides, macrolides, beta-lactams, nitrofuran, quinolones, and cephalosporins)

Response: We have reworded the sentence.

[Figure. 4 Antibiotic-resistant trends of the antibiotics. The antimicrobial resistance of the selected isolates to selected isolates showed that Escherichia coli and Salmonella spp., exhibited higher resistance to tetracyclines and macrolides across all samples. Whereas Klebsiella spp., alone showed susceptibility to nitrofurans and aminoglycosides, Cephalosporins and aminoglycosides were effective against most of the pathogens.]

Comment20: Line 284 in Figure 4 removes the following (Tetracyclines, Aminoglycosides, Macrolides, Beta-Lactams, Nitrofuran, Quinolones, and Cephalosporins) to Escherichia coli, Salmonella spp, Proteus spp, Klebsiella spp, Staphylococci spp, and Pseudomonas spp.

Response: We have removed it as suggested.

Comment21: Lines 287-293: Revise the paragraph and move into the discussion section “The influence of the COVID-19 pandemic on the livestock industry”, line 318

Response: We agree that some of the text should be moved to the discussion but have retained the interpretation of Figure 5.

Comment22: Line 295” in Figure would be better if add the cities name on Figures

Response: The affected cities’ names have been added to figure 1.

Comment23: Lines 297-301: Revise the paragraph and avoid repetition from the sentences from the Result section

Response: This has been revised

Comment24: Lines 304-308: Revise the sentence and avoid repetition from the sentences from the Result section

Response: We have made the correction as advised.

Comment25: Line 336; you have to Italicize all bacterial names (genus and species)

Response: We have taken note of that. This was addressed initially as suggested in comment12

Reviewer 2 Report

This paper is suitable for te pubication

Author Response

Comments and Suggestions for Authors: This is an interesting survey of AMR from livestock with specific geographical focus. The study seems straightforward. It would be nice to tie these results with genetics, but phenotypes are always the most convincing evidence.

Response: We thank the reviewer for their comment.  We agree with you that it would be nice to have some genetics added. However, the lab presently limits its diagnostic investigations to basic microbiological investigations involving biochemical tests and fermentation sugars. Further studies will be conducted when funded to look into this.

Comment1: The simple summary does not make the hypothesis tested or the overall finding obvious. You're examining the effects of antibiotics on what characteristic(s) of the bacteria? Or are you looking at effects on the farm animals themselves? It could be more clearly written, though I understand the challenge.

Response: This study was a retrospective audit that focused on observing how microorganisms from the diagnostic laboratory are distributed, and their antimicrobial susceptibility pattern over the years.  We stated in lines 105-109 that “This data audit covers a period of four years and aims to provide a snapshot of the microorganisms isolated at the bacteriology laboratory of the National Veterinary Research Institute (NVRI) Vom Nigeria. It also seeks to understand these pathogens’ distribution and antimicrobial resistance.”

Comment2: Revisit this sentence on line 126: All E. coli strains contain two and three plasmids that facilitate their 126 ability to acquire and distribute antimicrobial resistance horizontally to other microbes or 127 the environment.

Response: we have revised the statement as “Some E. coli strains may carry  plasmids that facilitate their ability to acquire and distribute antimicrobial resistance horizontally to other microbes or the environment.”

Comment3: Check syntax for sources cited throughout document.

Response: Thank you for that important observation. We have made extra efforts in reviewing the entire manuscript to match citations with their appropriate references.

Comment4: Check Table 1 for formatting.

Response: Table 1 has been revised.

Comment5: It would be nice to know how the bacterial isolate increase over the years compares with the number of samples submitted to this laboratory system.

Response: We quite agree that providing a ratio of isolates to samples entered in a particular year will be a nice way to present our data. We have expanded the Table 2. to include that.

Comment6: Figure 4 is nice. Regression analysis would be helpful to show trends over the years for the more variable data sets.

Response: Thank you for your comment. We agreed, and that is why we used regression analysis to estimate the % annual increase presented in Table 2.

Comment7: It would also be nice to discuss how this relates to clinical cases in humans if possible over the years.

Response: This is a good suggestion. Although we had mentioned this in the discussion, we have expanded the discussion on how humans could possibly be affected by a rising antimicrobial resistance in animals. Please find in the second paragraph of the discussion.

Reviewer 3 Report

This is an interesting survey of AMR from livestock with specific geographical focus. The study seems straightforward. It would be nice to tie these results with genetics, but phenotypes are always the most convincing evidence. 

General comments:

The simple summary does not make the hypothesis tested or the overall finding obvious. You're examining the effects of antibiotics on what characteristic(s) of the bacteria? Or are you looking at effects on the farm animals themselves? It could be more clearly written, though I understand the challenge.

Revisit this sentence on line 126: All E. coli strains contain two and three plasmids that facilitate their 126 ability to acquire and distribute antimicrobial resistance horizontally to other microbes or 127 the environment. 

Check syntax for sources cited throughout document.

Check Table 1 for formatting.

It would be nice to know how the bacterial isolate increase over the years compares with the number of samples submitted to this laboratory system. 

Figure 4 is nice. Regression analysis would be helpful to show trends over the years for the more variable data sets.

It would also be nice to discuss how this relates to clinical cases in humans if possible over the years.

The writing is good. Review the simple summary and strengthen it and review the paper for general syntax issues.

Author Response

(The authors gave the same response as above.)
